# Uncovering associations between pre-existing conditions and COVID-19 Severity: A polygenic risk score approach across three large biobanks

Lars G. Fritsche[1,2¤]*, Kisung Nam[3], Jiacong Du[1,2], Ritoban Kundu[1,2], Maxwell Salvatore[1,2], Xu Shi[1], Seunggeun Lee[3], Stephen Burgess[4,5], Bhramar Mukherjee[1,2,6]*

1 Department of Biostatistics, University of Michigan School of Public Health, Ann Arbor, Michigan, United States of America, 2 Center for Precision Health Data Science, University of Michigan School of Public Health, Ann Arbor, Michigan, United States of America, 3 Graduate School of Data Science, Seoul National University, Seoul, South Korea, 4 MRC Biostatistics Unit, University of Cambridge, Cambridge, United Kingdom, 5 Cardiovascular Epidemiology Unit, University of Cambridge, Cambridge, United Kingdom, 6 Michigan Institute for Data Science, University of Michigan, Ann Arbor, Michigan, United States of America

¤ Current address: Department of Biostatistics, School of Public Health University of Michigan, 1415 Washington Heights, SPH Tower Room 4633, Ann Arbor, Michigan 48109, United States of America
* larsf@umich.edu (LGF); bhramar@umich.edu (BM)

## Abstract

### Objective

To overcome the limitations associated with the collection and curation of COVID-19 outcome data in biobanks, this study proposes the use of polygenic risk scores (PRS) as reliable proxies of COVID-19 severity across three large biobanks: the Michigan Genomics Initiative (MGI), UK Biobank (UKB), and NIH All of Us. The goal is to identify associations between pre-existing conditions and COVID-19 severity.

### Methods

Drawing on a sample of more than 500,000 individuals from the three biobanks, we conducted a phenome-wide association study (PheWAS) to identify associations between a PRS for COVID-19 severity, derived from a genome-wide association study on COVID-19 hospitalization, and clinical pre-existing, pre-pandemic phenotypes. We performed cohort-specific PRS PheWAS and a subsequent fixed-effects meta-analysis.

### Results

The current study uncovered 23 pre-existing conditions significantly associated with the COVID-19 severity PRS in cohort-specific analyses, of which 21 were observed in the UKB cohort and two in the MGI cohort. The meta-analysis yielded 27 significant phenotypes predominantly related to obesity, metabolic disorders, and cardiovascular conditions. After adjusting for body mass index, several clinical phenotypes, such as hypercholesterolemia

**Data Availability Statement:** We confirm that the underlying numerical data for all graphs and

summary statistics has been provided as Supporting Information.

**Funding:** LGF is supported by the University of Michigan Precision Health (grant number U063790). LGF and XS are also supported by the University of Michigan Precision Health (grant number U075994). SB is supported by the Wellcome Trust (grant number 225790/Z/22/Z) [https://wellcome.org/] and the United Kingdom Research and Innovation Medical Research Council (grant number MC_UU_00002/7) [https://www.ukri.org/councils/mrc/]. SL is supported by the Brain Pool Plus (BP+) Program through the National Research Foundation of Korea (NRF), funded by the Ministry of Science and ICT (grant number 2020H1D3A2A03100666) [https://www.nrf.re.kr/eng/main]. BM is supported by the National Cancer Institute (grant number CA 046592) [https://www.cancer.gov/]. The funders had no role in the study design, data collection and analysis, decision to publish, or preparation of the manuscript.

**Competing interests:** I have read the journal's policy and the authors of this manuscript have the following competing interests: LGF is a Without Compensation (WOC) employee at the VA Ann Arbor, a United States government facility. SB is a paid statistical reviewer for PLOS Medicine.

and gastrointestinal disorders, remained associated with an increased risk of hospitalization following COVID-19 infection.

## Conclusion

By employing PRS as a proxy for COVID-19 severity, we corroborated known risk factors and identified novel associations between pre-existing clinical phenotypes and COVID-19 severity. Our study highlights the potential value of using PRS when actual outcome data may be limited or inadequate for robust analyses.

## Author summary

In our study, we addressed a pressing issue arising from the COVID-19 pandemic, namely identifying who is vulnerable to severe complications that require hospitalization. Some pre-existing health conditions can increase risk for hospitalization, but identifying these conditions is challenging when directly relying on imperfect health records. Instead, we used genetic information to predict COVID-19 severity. By combining the risk effects of multiple genetic variants, as estimated by an external study, into a single score, we aimed to capture the predisposition to severe illness from COVID-19.

We analyzed genetic data from over half a million individuals in three large biobanks and discovered connections between this genetic risk score and specific pre-existing conditions, including obesity and heart diseases. Our approach provides a way to overcome challenges in data availability and quality and offers valuable insights, even when actual COVID-19 outcomes were inconsistently or not documented. Our work lays a foundation for a better understanding of individuals at risk and emphasizes how genetics can inform public health decisions and personalized care. Ultimately, our approach showcases how to obtain key information on managing risks for not just COVID-19 but also other infectious diseases.

## Introduction

The Coronavirus Disease-2019 (COVID-19) pandemic has posed unprecedented challenges to public health and healthcare systems worldwide. Emerging evidence suggests that certain pre-existing conditions and genetic factors play a crucial role in determining COVID-19 severity, making it essential to explore these associations further. Precision health has emerged as a promising approach to personalize interventions and improve health outcomes [1]. One emerging aspect of precision health is using polygenic risk scores (PRS) for risk stratification and identifying individuals susceptible to various diseases [2–5].

Tracking the severity of COVID-19 within the context of biobanks presents distinct challenges, such as underreporting, delayed data releases, or lack of approved diagnosis codes [6–9]. Generating and using a PRS as a proxy for COVID-19 severity would provide two major advantages over studies that directly used electronic health records (EHR)-based hospitalization. First, a PRS would be available for every individual of a biobank cohort with genotype data, thus avoiding smaller complete-case analyses. For example, in the current study of three biobanks, each with EHR each spanning between two and three years of the COVID-19 pandemic, severe COVID-19 cases were consistently below 1%. In contrast, the analysis with a

continuous PRS spanned more than 500,000 individuals underscoring the potential limitation of relying solely on direct observations. Second, using a PRS can make the analysis less prone to selection bias (e.g., who gets COVID outcome data recorded in EHR and who does not) and confounding (health care seeking behavior and access factors), with genetic variants proven to address potential confounding effectively in Mendelian randomization studies [10].

Consequently, a PRS might enable a more accurate investigation of associations between COVID-19 severity and pre-existing conditions. Even though the clinical applicability of PRSs is still nascent, these scores can establish robust associations despite capturing only a smaller fraction of a trait's variability. For example, even with limited sample sizes of measured outcomes in biobanks, PRSs can provide valuable insights into the genetic control of biological pathways and disease associations [11].

Most existing studies exploring the link between pre-existing conditions and COVID-19 severity focus primarily on the population tested for COVID-19 [12,13]. Such analyses are susceptible to confounding due to potential testing bias and misclassification error [14–18]. The issue of testing bias is further complicated by testing resources that vary across time, healthcare systems, and countries. Additionally, the definition of severe COVID-19 is often related to hospitalization or intensive care unit (ICU) admission or respiratory failure and depends on the health practices [19,20]. Using a PRS as a proxy for COVID-19 severity, as opposed to the actual recorded COVID-19 severity diagnosis, may reduce bias in association testing, as it was assigned at birth and is thus unlikely to be associated with the likelihood of being tested. Also, a PRS would be available for every individual of a genotyped biobank cohort, reducing selection bias. Moreover, using a PRS may indicate shared genetic susceptibility between COVID-19 and other pre-existing conditions [21].

The significance of genetic factors in the susceptibility and severity of COVID-19 has been demonstrated in recent studies [22,23], emphasizing the need to understand the interplay between genetics and pre-existing conditions. For instance, an elevated plasma level of 2'-5'-oligoadenylate synthetase 1 (OAS1), involved in viral clearance, has been found to be predictive against severe COVID-19 [23,24] and influenced by a common haplotype of *OAS1* [25]. In addition, variants in the tyrosine kinase 2 (*TYK2*) gene [23], in the Toll-like receptor 7 (*TLR7*) gene [26], and the interferon-alpha/beta receptor 2 (*IFNAR2*) gene [19,23] which play crucial roles in immune response regulation, pathogen recognition pathways, and interferon signaling respectively, have been reported to be linked to severe COVID-19 outcomes. Further, a meta-analysis revealed significant associations between certain genetic polymorphisms in the renin-angiotensin-aldosterone system (RAAS)-related genes and COVID-19 susceptibility and severity [27]. Another systematic review and meta-analysis revealed that genetic variants within the angiotensin-converting enzyme 1 and 2 genes (*ACE* and *ACE2*, respectively) and the transmembrane serine protease 2 (*TMPRSS2*) gene were associated with COVID-19 severity [28]. Moreover, a systematic review found that *ACE* and the interferon-induced transmembrane protein 3 (*IFTM3*) gene polymorphisms may lead to a genetic predisposition for severe lung injury in COVID-19 patients [29]. Also, genome-wide association study (GWAS) meta-analyses of the COVID-19 Host Genetics Initiative identified 14 loci associated with COVID-19 disease severity or hospitalization [30]. These findings point to a genetic predisposition for severe COVID-19 that could be assessed and utilized through a PRS, as previously suggested [31–33].

Building on the existing knowledge of genetic factors influencing COVID-19 severity and our previous work on COVID-19 outcomes [22,23,34], health-related exposure trait PRS [35], and PRS for cancer traits [36], our study aims to employ PRS-CS, a polygenic prediction method, and summary statistics from a GWAS on hospitalization of the COVID-19 Host Genetics Initiative to develop a genome-wide PRS for COVID-19 severity [37]. This approach

seeks to address the limitations of conventional risk assessments, which are prone to biases and inconsistencies across health systems and biobanks [38–41].

This study uses data from three biobanks: the Michigan Genomics Initiative (MGI) [42], the UK Biobank (UKB) [43], and NIH's relatively new All of Us cohort [44] to examine associations between a newly developed COVID-19 severity PRS and pre-existing conditions via phenome-wide association studies (PheWAS). The objective is to identify phenotypes potentially linked to an increased risk of severe COVID-19 outcomes. We acknowledge the previously reported challenges regarding the transferability of European-ancestry (EUR)-based PRSs [45] yet also explored this PRS in non-European ancestry groups. Our analyses were stratified by biobanks and inferred ancestry due to varying sampling strategies in these three biobanks. While we strive for inclusivity, we primarily report on the well-powered analyses for the EUR samples, both stratified and meta-analyses.

Our approach aims to inform targeted interventions and prevention measures and to shed light on shared genetic susceptibility of COVID-19 severity and pre-existing conditions. The implications of our findings extend beyond the current pandemic context, potentially contributing to more effective strategies for managing individuals with pre-existing conditions who are at high risk for severe outcomes from other infectious diseases. Given the availability of genetic data in EHR-linked biobanks or via commercial genetic testing services, a significant number of individuals could benefit from such advancements, marking a critical step towards leveraging genetic data for personalized healthcare delivery and public health planning.

## Subjects and methods

### Cohorts

**Ethics statement.** Data collection adhered to the Declaration of Helsinki principles. The University of Michigan Medical School Institutional Review Board reviewed and approved the consent forms and protocols of MGI study participants (IRB ID HUM00099605 and HUM00155849). Opt-in written informed consent was obtained. UK Biobank received ethical approval from the NHS National Research Ethics Service North West (11/NW/0382). The All of Us research program received ethical approval from the All of Us IRB (Protocol Title: All of Us Research Program; Sponsor: National Institutes of Health (NIH); Protocol Version: Protocol v1.19p, Operational Protocol; IRB Approval Date: December 3, 2021).

**Michigan Genomics Initiative (MGI).** Adult participants aged between 18 and 101 years at enrollment were recruited through the Michigan Medicine health system between 2012 and 2020. Participants have consented to allow research on their biospecimens and EHR data and linking their EHR data to national data sources such as medical and pharmaceutical claims data. The data used in this study included diagnoses coded with the Ninth and Tenth Revision of the International Statistical Classification of Diseases (ICD9 and ICD10) with clinical modifications (ICD9-CM and ICD10-CM), body mass index [BMI], genetically inferred sex, pre-computed principal components (PCs), kinship estimates down to the third degree, genotyping batch, recruitment study, age, and imputed genotype data. A comprehensive description of the cohort and its linked data sources has been previously reported [42]. Further details about MGI are available online (see Web Resources).

DNA from 60,052 blood samples was genotyped on customized Illumina Infinium CoreExome-24 bead arrays and subjected to various quality control filters, resulting in 502,255 polymorphic variants. Principal components and European / non-European ancestry were estimated by projecting all genotyped samples into the space of the principal components of the Human Genome Diversity Project reference panel using the software PLINK (938 individuals) [46,47]. We assessed pairwise kinship with the software KING [48] and used the software

FastIndep to reduce the cohort to a maximal subset that contained no pairs of individuals with $3^{rd}$- or closer-degree relationship [49] (no pairwise kinship coefficient > 0.08838). Additional genotypes were obtained using the Haplotype Reference Consortium reference panel of the Michigan Imputation Server [50] and included imputed variants with $R^2 \geq 0.3$ and minor allele frequency (MAF) $\geq 0.01\%$.

We estimate the fraction of each MGI participant's genome originating from African (AFR), Admixed American (AMR), East Asian (EAS), European (EUR), and South Asian (SAS) ancestry using ADMIXTURE (v1.3.0) [51] and the HGDP reference [52]. We assigned ancestry to individuals with an ancestry fraction greater than 70% while defining the remaining samples as having 'other' ancestry.

**UK Biobank (UKB).** UKB is a population-based cohort collected from multiple sites across the United Kingdom and includes over 500,000 participants aged between 40 and 69 years when recruited in 2006–2010 [43]. The open-access UK Biobank data used in this study included ICD9 and ICD10 codes, age, BMI, genetically inferred sex, PCs, precomputed inferred ancestry, precomputed pair-wise kinship coefficients, array-based and HRC-imputed genotyping data. We used the UK Biobank Imputed Dataset (v3) and limited analyses to variants with imputation information score >= 0.3 and MAF $\geq 0.01\%$. We used the software FastIndep to reduce the cohort to a maximal subset that contained no pairs of individuals with a $3^{rd}$- or closer-degree relationship [49] (no pairwise kinship coefficient > 0.08838). For the ancestry prediction, we applied Online Augmentation, Decomposition and Procrustes (OADP) method to the genotype data with 2492 samples from the 1000 Genomes Project data as the reference (FRAPOSA; see **Web Resources**) [53] to infer the super population's membership (AFR: African, AMR: Admixed American, EAS: East Asian, EUR: European, and SAS: South Asian ancestry).

**All of Us cohort.** All of Us is a population-based research study that enrolls adults from multiple sites across the USA. As of June 6, 2022, "All of Us Controlled Tier Dataset v6" included over 372,380 participants with linked data available, 227,740 had EHR data, and 98,590 had whole genome sequencing data available (see **Web Resources**). For the current study, we used available ICD9-CM and ICD10-CM codes, age, BMI, genetically inferred sex, precomputed principal components (PCs), precomputed inferred ancestry, a pre-computed set of unrelated individuals (pair-wise kinship coefficient < 0.1), as well as pre-processed whole genome sequencing data.

We used the Controlled Tier Dataset (version 6, n = 98,590) and used the provided auxiliary data to remove 156 samples that failed quality control filters and 4,069 related individuals (no pairwise kinship coefficient > 0.1) to obtain a set of 94,377 unrelated individuals.

We assigned ancestry to individuals with a precomputed inferred ancestry fraction greater than 70% while defining the remaining samples as having 'other' ancestry. This threshold, while arbitrary, was selected to strike a balance between preserving sample size and increasing homogeneity in each ancestry group.

## COVID-19 Severity Polygenic Risk Score (PRS) Construction

A PRS combines information across a defined set of genetic loci, incorporating each locus's association with the target trait. The PRS for person $j$ takes the form

$PRS_j = \sum_i \hat{\beta}_i G_{ij}$ where $i$ indexes the included loci for that trait, weight $\hat{\beta}_i$ is the estimated log odds ratio retrieved from the external GWAS summary statistics for locus $i$, and $G_{ij}$ is a continuous version of the measured dosage data for the risk allele on locus $i$ in subject $j$.

We downloaded the GWAS meta-analysis summary statistics on COVID-19 severity from the COVID-19 Host Genetics Initiative (COVID19-hg GWAS meta-analyses round 7; release

date: April 8, 2022; also see **Web Resources**). We considered summary statistics from two GWAS meta-analyses: (1) "B1_ALL": hospitalized COVID-19 versus not hospitalized COVID-19 ("B1_ALL_leave_23andme" [16512 cases vs. 71321 controls] and (2) "B2_ALL": hospitalized COVID-19 versus population controls ("B2_ALL_leave_23andme" [44,986 cases vs. 2,356,386 controls]. To mitigate the risk of overfitting and to ensure the robustness of our findings, PRSs for the UK Biobank cohort were specifically calculated using GWAS meta-analysis results that excluded UK Biobank samples ('leave_23andme_and_UKBB'): "B1_ALL_leave_23andme_and_UKBB" [12,455 cases vs. 61,144 controls]) and "B2_ALL_leave_23andme_and_UKBB" [40,929 cases vs. 1,924,400 controls]). In contrast, the PRS for the other two cohorts were based on GWAS that included UK Biobank samples. The underlying meta-analyses utilized a standard association model, including covariates for age, sex, the first 20 principal components (PCs), and study-specific technical covariates, excluding heritable risk factors and comorbidities. Each contributing cohort conducted GWAS under this framework, employing the SAIGE software [54] to account for relatedness and case-control imbalance. For a comprehensive account of the participant demographics and individual study contributions, see S1 Table, which lists sample sizes and ancestry data for the "B1_ALL" meta-analysis.".

In the underlying "B1_ALL" GWAS, COVID-19 severity was defined based on hospitalization due to COVID-19-related symptoms (cases) and non-hospitalization 21 days after the test (controls), both with laboratory-confirmed SARS-CoV-2 infection (RNA and/or serology based). This set of summary statistics may eliminate testing bias but may not be generalizable. For the underlying "B2_ALL" GWAS, population-based controls (non-cases) were selected. This set of summary statistics may be more generalizable with population-based controls that may include a mix of tested and untested individuals and asymptomatic and mildly symptomatic cases of COVID-19 in the control group. While not specified, hospitalized cases may include deceased individuals.

We used the software package "PRS-CS" [37] to define PRS weights based on a Bayesian regression framework employing continuous shrinkage (CS) priors. Briefly, PRS-CS adjusts the SNPs 'effect sizes to account for their associations with the trait of interest and the local LD patterns, thereby resulting in a PRS that more accurately reflects the complex genetic architecture of a trait. It does not require individual-level data but integrates GWAS summary statistics with a provided, precomputed, ancestry-specific LD reference panel for up to 1,117,425 common, non-ambiguous, autosomal SNPs based on samples of the UK Biobank (see **Web Resources**).

We opted for PRS-CS because it has demonstrated superior performance to other PRS methods, likely attributable to its adaptable modeling assumptions [55]. We only included autosomal variants that overlap between the GWAS summary statistics, LD reference panel, and the target cohort (MGI: 1,113,665 variants; UK: 1,116,734 variants; or All of Us: 1,116,233 variants). A full list of weights can be downloaded from our website (see **Web Resources**). We used PLINK 2.00 alpha (see **Web Resources**) and the PRS-CS-derived weights to calculate the dosage-based COVID-19 severity PRS for each individual of each target dataset. Finally, we centered the PRS of each target dataset and ancestry group to a mean of 0 and scaled it to a standard deviation (SD) of 1.

**Covariates.** Similar to our previous COVID-19 studies [13,34,56–58], we considered the following key covariates to control for potential confounding factors. Age at the time of recruitment (UKB) or last observed visit (All of Us, MGI), whichever came last, was considered a continuous variable for each participant. In addition, genetically inferred sex-at-birth was incorporated as a binary variable. The first four genetic principal components were included in the analysis to address population stratification and ancestry differences among the participants. Elixhauser comorbidity scores were calculated using the R package

"comorbidity" and the AHRQ weighting scheme [59] to account for the participants' overall health status prior to the COVID-19 pandemic. For the MGI and UKB cohorts, the genotyping array was also considered a covariate to control for potential biases stemming from differences in genotyping technologies. In some analyses, we additionally adjusted for body mass index (BMI), defined as the median of all observations.

**Phenome generation.** For all three datasets, MGI, UKB, and All of Us, we used ICD codes (MGI and All of Us: ICD9-CM and ICD10-CM; UKB: ICD9 and ICD10 codes) recorded before the COVID-19 pandemic (MGI: last entry February 2020, UKB: last entry from March 2017, and All of Us: filtered to entries before March 11, 2020) and aggregated these pre-COVID-19 pandemic ICD codes into up to 1817 PheWAS codes (PheCodes) using the PheWAS R package (Version: 0.99.5-4, further details are described elsewhere [60]). We limited the phenomes of an ancestry group to case-control studies with >50 cases (**S3–S8** Tables).

**COVID-19 outcome data.** We extracted data on hospitalized COVID-19 cases from each cohort to determine their distribution across the PRS deciles. For UKB, we identified individuals who, as of February 2023 (the data extraction date), had in-patient data with a COVID-19 diagnosis defined by the ICD10 codes U07.1 and U07.2. Additionally, we also considered individuals who self-reported being hospitalized with COVID-19. For MGI, we considered individuals who, as of March 2023, had an admitting diagnosis code of U07.1 or U07.2 or who had an observed COVID-19 diagnosis (U07.1 or U07.2) within 7 days before or 30 days after being admitted to the hospital. For the All of Us cohort, we identified individuals who were hospitalized with COVID-19, as of January 1, 2022 (the data cutoff date of the All of Us Controlled Tier Dataset v6), by extracting the condition concept for COVID-19 (840539006) and defined hospitalizations as inpatient and intensive care visits.

**Phenome-wide Association Studies (PheWAS).** To identify phenotypes associated with the COVID-19 severity PRS, we conducted Firth bias-corrected logistic regression. This method addresses potential issues of small sample sizes, rare events, and separation issues in logistic regression models. Accordingly, we fit the following model for each PheCode of the pre-COVID-19 phenomes.

$$
\begin{aligned}
logit&(P(PheCode = 1|PRS, Covariates)) \\
&= \beta_0 + \beta_{PRS}PRS + \beta_{age}age + \beta_{sex}sex + \beta_{PC1}PC1 + \beta_{PC2}PC2 + \beta_{PC3}PC3 + \beta_{PC4}PC4 \\
&\quad + \beta_{Comobidity}Comorbidity + \beta_{array}Array,
\end{aligned}
\tag{1}
$$

where covariates were age, sex, the first four genetic principal components obtained from the principal component analysis, pre-COVID-19 Elixhauser Comorbidity Score (AHRQ), and the genotyping array (only for MGI and UKB). We performed PheWAS with and without adjustment for BMI (also see "Covariates" above). For a given phecode, PheWAS results correspond to Wald tests corresponding to $H_0$: $\beta_{PRS} = 0$. vs $H_0$: $\beta_{PRS} \neq 0$ across the phenome.

We performed meta-analyses of the PheWAS results of the three cohorts by ancestry group using a fixed-effect model implemented in the R package "meta" [61]. In Manhattan plots, we present –log10 (p-value) corresponding to tests for the association of the underlying phenotype with the COVID-19 severity PRS. Directional triangles on the PheWAS plot indicate whether a phenotype was positively (pointing up) or negatively (pointing down) associated with the COVID-19 severity PRS. We applied the Bonferroni correction to account for multiple hypothesis testing in our PheWAS by dividing the desired family-wise error rate (α = 0.05) by the total number of tests conducted.

### Artificial Intelligence tools

During the preparation of this manuscript, artificial intelligence (AI) tools were utilized as follows: (1) Manuscript Revision: tools such as Grammarly and ChatGPT-4 were employed to revise the manuscript text. This involved refining grammar, correcting typos, improving the flow and clarity of the content, and optimizing text length. (2) Coding Debugging and Troubleshooting: For the software coding aspects of this study, GitHub Copilot and ChatGPT-4 were used to troubleshoot coding issues. Noteworthy, the core ideas, hypotheses, interpretations, results, conclusions, limitations, and implications of the study remain entirely the original work and views of the listed authors. All content was meticulously reviewed to ensure accuracy, originality, and to prevent potential issues related to plagiarism. Relevant sources have been cited accordingly.

## Results

We present the results of our analyses, which primarily include descriptive characteristics of the three biobanks, PheWAS results for each biobank, a meta-analysis across the biobanks, and additional PheWAS analyses for African and East Asian ancestry groups. Our findings reveal significant associations between the COVID-19 Severity PRS and pre-existing conditions, with notable differences across cohorts and ancestries.

### Descriptive characteristics of the three biobanks

We first offer a detailed account of the demographic and health parameters of the three biobanks that form the basis of our analysis. Recognizing the similarities and differences between these cohorts is crucial as they provide valuable context for interpreting our findings and allow for a more nuanced appreciation of the associations between the COVID-19 Severity PRS (a genetic predisposition to COVID-19 severity) and pre-existing conditions. **Table 1** provides the characteristics of the three pre-COVID-19 cohorts with genotype data, namely MGI (n=60,052), UKB (n=485,442), and All of Us (n=98,398). Upon closer inspection, it is apparent that demographic and health parameters vary across cohorts. The mean age ranges from 55.32 years in All of Us to 60.82 years in UKB, while the distributions across age categories differ, too. The percentage of closely related individuals was comparable across cohorts, with the highest in UKB (7.1%) and the lowest in All of Us (4.9%).

The UKB cohort is predominantly of European ancestry (94.6%), reflecting a large proportion of Non-Hispanic White participants (94.3%). In contrast and by design, the All of Us cohort displays greater diversity in ancestry and race/ethnicity. For example, 23.8% of the cohort has African ancestry, and 19.7% identify as Hispanic/Latino.

Further, **Table 1** shows that the mean BMI varies across cohorts, with MGI reporting the highest value (29.90) and the largest proportion of obese participants (41.9%). Substantial variations regarding health characteristics from the EHR can also be observed, including age at the first and last recorded diagnosis, Elixhauser AHRQ scores, and the number of unique PheCodes per person. The mean age at the first diagnosis was 45.98 (SD: 17.40) for MGI, 52.40 (SD: 10.38) for UKB, and 44.81 (SD: 16.66) for All of Us. Likewise, the age at the most recent recorded diagnosis diverged across cohorts, with MGI registering a mean at 57.89 (SD: 16.86), UKB at 60.65 (SD: 10.33), and All of Us at 54.24 (SD: 17.21). Consequently, there was substantial variation regarding the time in the EHR, i.e., the time between the age at the first and last recorded diagnosis, with MGI averaging at 11.91 years, UKB at 8.25 years, and All of Us at 9.43 years.

MGI had the highest mean Elixhauser AHRQ score (7.93), while UKB and All of Us had markedly lower mean scores (1.64 and 1.60, respectively). These lower scores might be driven

**Table 1. Characteristics of the three cohorts with genotype data, pre-COVID-19 datasets\*.**

| | MGI | UKB | All of Us |
|---|---|---|---|
| n | 60052 | 485442 | 98398 |
| Age | | | |
| mean (SD) | 57.89 (16.86) | 60.82 (9.22) | 55.32 (16.88) |
| categories, n (%) | | | |
| [18,35) | 7352 (12.2) | 0 (0.0) | 15670 (15.9) |
| [35,50) | 11245 (18.7) | 74370 (15.3) | 21630 (22.0) |
| [50,65) | 18418 (30.7) | 227255 (46.8) | 29117 (29.6) |
| [65,80) | 18294 (30.5) | 183804 (37.9) | 26466 (26.9) |
| >=80 | 4743 (7.9) | <20 (<0.005) | 5515 (5.6) |
| Female (%) | 28176 (46.9) | 263372 (54.3) | 59569 (60.5) |
| Closely related\*\* (%) | 3694 (6.2) | 34587 (7.1) | 4778 (4.9) |
| Ancestry (%) | | | |
| European | 50283 (83.7) | 459284 (94.6) | 49019 (49.8) |
| African | 3036 (5.1) | 8101 (1.7) | 23454 (23.8) |
| East Asian | 1112 (1.9) | 2524 (0.5) | 2122 (2.2) |
| Central/South Asian | 770 (1.3) | 10053 (2.1) | 995 (1.0) |
| Admixed American | 28 (0.0) | 0 (0.0) | 15527 (15.8) |
| West Asian | 330 (0.5) | 0 (0.0) | 204 (0.2) |
| Other | 4493 (7.5) | 5480 (1.1) | 7077 (7.2) |
| Race/Ethnicity (%) | | | |
| Non-Hispanic White | 51177 (85.2) | 457872 (94.3) | 49561 (50.4) |
| Non-Hispanic Black/African American | 3752 (6.2) | 7623 (1.6) | 21207 (21.6) |
| Non-Hispanic Asian | 1770 (2.9) | 10884 (2.2) | 2959 (3.0) |
| Hispanic/Latino | 848 (1.4) | 0 (0.0) | 19354 (19.7) |
| Other | 1573 (2.6) | 7240 (1.5) | 3051 (3.1) |
| n/a | 932 (1.6) | 1823 (0.4) | 2266 (2.3) |
| BMI | | | |
| mean (SD) | 29.90 (7.24) | 27.42 (4.79) | 29.67 (7.51) |
| categories, n (%) | | | |
| Underweight (< 18.5) | 619 (1.0) | 2498 (0.5) | 1222 (1.2) |
| Healthy weight [18.5, 25) | 14820 (24.7) | 157570 (32.5) | 26459 (26.9) |
| Overweight [25, 30) | 19388 (32.3) | 205790 (42.4) | 29985 (30.5) |
| Obese (>= 30) | 25172 (41.9) | 117737 (24.3) | 39313 (40.0) |
| n/a | 53 (0.1) | 1847 (0.4) | 1419 (1.4) |
| Individuals without Recorded Diagnoses, n (%) | 0 (0) | 85522 (17.6) | 29026 (29.5) |
| Age at first recorded diagnosis, mean (SD) | 45.98 (17.40) | 52.40 (10.38) | 44.81 (16.66) |
| Age at last recorded diagnosis, mean (SD) | 57.89 (16.86) | 60.65 (10.33) | 54.24 (17.21) |
| Elixhauser AHRQ, mean (SD) | 7.93 (13.99) | 1.64 (5.28) | 1.60 (8.05) |
| Elixhauser AHRQ binned, n (%) | | | |
| <0 | 16185 (27.0) | 80459 (16.6) | 23604 (24.0) |
| 0 | 8133 (13.5) | 289971 (59.7) | 48802 (49.6) |
| 1-4 | 6521 (10.9) | 35837 (7.4) | 7675 (7.8) |
| >=5 | 29213 (48.6) | 79175 (16.3) | 18317 (18.6) |
| Unique PheCodes per Person, mean (SD) | 67.72 (60.42) | 8.84 (11.29) | 26.24 (37.91) |

(*Continued*)

**Table 1.** (Continued)

|  | MGI | UKB | All of Us |
| --- | --- | --- | --- |
| Hospitalized with COVID-19, n (%)*** | 307 (0.51) | 3165 (0.65) | 703 (0.71) |

\* MGI: data from before 2020-02-13; All of Us: data from before 2020-03-11; UK Biobank: data from before 2018 (last observed date March 2017)

\*\* MGI, UKB: pair-wise kinship coefficient >= 0.0883, All of Us: pair-wise kinship coefficient > 0.1

\*\*\* MGI: COVID-19 data from before March 2023; All of Us: data from before 2022-01-01; UK Biobank: data from before February 2023.

Abbreviations: AHRQ, Agency for Healthcare Research and Quality; MGI, Michigan Genomics Initiative; SD, standard deviation; UKB, UK Biobank

by their larger proportion of individuals without any recorded diagnoses (MGI: 0%, UKB: 17.6%, and All of Us: 29.5%). Similarly, the mean number of unique clinical phenotypes, as represented by unique PheCodes per person, varied considerably among the cohorts. MGI revealed an average of 67.72 unique phecodes (SD: 60.42), the highest value, in contrast to UKB with 8.84 (SD: 11.29) and All of Us with 26.24 unique phecodes (SD: 37.91). The notable differences in EHR length and the mean number of measurements further underscored the dissimilarities in overall health status between the hospital-based MGI and the two population-based cohorts, UKB and All of Us. In the three cohorts, the individuals hospitalized due to COVID-19 were consistently below 1%: MGI, 307 (0.51%); UKB, 3,165 (0.65%); and All of Us, 703 (0.71%).

Given that we will use a COVID-19 severity PRS derived from predominantly inferred European ancestry (EUR) cohorts, we next limited the analytical datasets to unrelated individuals of inferred European ancestry (MGI: n=47,257, UKB: n=425,787, and All of Us: n=47,401). The resulting relative sample size loss was 51.8% for All of Us, 21.3% for MGI, and 12.3% for UKB, reflecting the varying cohort diversity. **Table 2** presents the corresponding summary statistics for the same three cohorts as in **Table 1**. Both tables display similar patterns, such as age and age distribution, mean BMI, and Elixhauser comorbidity scores. However, a notable difference can be observed in the mean number of unique PheCodes per person for the All of Us cohort, which increased from 26.24 in **Table 1** to 31.89 in **Table 2**. Since **Table 2** only includes EUR individuals, this change suggested prevailing disparities in healthcare access between non-EUR and EUR individuals. The proportion of individuals hospitalized due to COVID-19 were slightly lower in the analytical dataset, yet very comparable: MGI, 217 (0.46%); UKB, 2,560 (0.60%); and All of Us, 298 (0.63%).

## PheWAS results in the three biobanks

We conducted biobank-specific PheWAS using the COVID-19 Severity PRS and the pre-pandemic phenomes of the analytical datasets (unrelated EUR ancestry groups, Table 2). Here, we focus on the PRS based on the hospitalized COVID-19 versus not hospitalized COVID-19 GWAS ("B1_ALL") of the COVID-19 Host Genetics Initiative because it is conditional on a laboratory-confirmed SARS-CoV-2 infection.

Our findings revealed 23 phenome-wide significant associations in MGI ($P < 3.6 \times 10^{-5}$) and UKB ($P < 3.6 \times 10^{-5}$), while none was found in the All of Us cohort (n = 47401). Most of these associations (21 out of 23) were observed in the UKB cohort (n = 425787), while two were identified in the MGI cohort (n = 47257, **Fig 1**).

In the UK Biobank cohort, we identified 21 significant associations involving 14 unique phecode groupings across six categories. These hierarchical groupings consist of a broader

**Table 2. Characteristics of the three analytical (unrelated, European ancestry) datasets*.**

|  | MGI | UKB | All of Us |
|---|---|---|---|
| n | 47257 | 425787 | 47401 |
| Age |  |  |  |
| mean (SD) | 59.18 (16.44) | 61.11 (9.09) | 59.26 (16.73) |
| categories, n (%) |  |  |  |
| [18,35] | 4827 (10.2) | 0 (0.0) | 5384 (11.4) |
| [35,50] | 8337 (17.6) | 60461 (14.2) | 8736 (18.4) |
| [50,65] | 14638 (31.0) | 200090 (47.0) | 12624 (26.6) |
| [65,80] | 15454 (32.7) | 165223 (38.8) | 16814 (35.5) |
| >=80 | 4001 (8.5) | <20 (<0.005) | 3843 (8.1) |
| Female (%) | 22398 (47.4) | 229753 (54.0) | 28608 (60.4) |
| BMI |  |  |  |
| mean (SD) | 29.95 (7.09) | 27.40 (4.77) | 29.10 (7.07) |
| categories, n (%) |  |  |  |
| Underweight (< 18.5) | 505 (1.1) | 2180 (0.5) | 514 (1.1) |
| Healthy weight [18.5, 25) | 11403 (24.1) | 138770 (32.6) | 13782 (29.1) |
| Overweight [25, 30) | 15282 (32.3) | 180589 (42.4) | 15171 (32.0) |
| Obese (>= 30) | 20041 (42.4) | 102929 (24.2) | 17221 (36.3) |
| n/a | 26 (0.1) | 1319 (0.3) | 713 (1.5) |
| Individuals without Recorded Diagnoses, n (%) | 0 (0) | 74836 (17.6) | 10859 (22.9) |
| Age at first recorded diagnosis, mean (SD) | 47.57 (17.04) | 52.64 (10.34) | 47.71 (16.52) |
| Age at last recorded diagnosis, mean (SD) | 59.18 (16.44) | 60.91 (10.23) | 58.00 (16.74) |
| Elixhauser AHRQ, mean (SD) | 8.42 (14.11) | 1.67 (5.30) | 2.29 (8.57) |
| Elixhauser AHRQ binned, n (%) |  |  |  |
| <0 | 12245 (25.9) | 69767 (16.4) | 11606 (24.5) |
| 0 | 5931 (12.6) | 254344 (59.7) | 20573 (43.4) |
| 1-4 | 5045 (10.7) | 31309 (7.4) | 4281 (9.0) |
| >=5 | 24036 (50.9) | 70367 (16.5) | 10941 (23.1) |
| Number of unique PheCodes per Person, mean (SD) | 67.09 (59.74) | 8.83 (11.25) | 31.89 (40.22) |
| Hospitalized with COVID-19, n (%)** | 217 (0.46) | 2560 (0.60) | 298 (0.63) |

* MGI: data from before 2020-02-13; All of Us: data from before 2020-03-11; UK Biobank: data from before 2018 (last observed date March 2017)

** MGI: COVID-19 data from before March 2023; All of Us: data from before 2022-01-01; UK Biobank: data from before February 2023.

Abbreviations: AHRQ, Agency for Healthcare Research and Quality; MGI, Michigan Genomics Initiative; SD, standard deviation; UKB, UK Biobank

parent phecode and their more specific child phecode(s). They include endocrine/metabolic disorders (diabetes mellitus [P = 4.48x10⁻⁶], hypercholesterolemia [P = 3.06x10⁻⁸], and obesity [P = 1.20x10⁻⁷]), mental disorders (alcohol-related disorders [P = 3.48x10⁻⁶], tobacco use disorder [P = 1.69x10⁻⁶], and substance addiction and disorders [P = 2.03x10⁻⁵]), cardiovascular conditions (hypertension [P = 2.13x10⁻⁸], essential hypertension [P = 2.86x10⁻⁸]), gastrointestinal disorders (esophageal conditions [P = 2.75x10⁻⁶], gastritis and duodenitis [P = 1.24x10⁻⁶], abdominal hernia [P = 6.06x10⁻⁶], functional digestive disorders [P = 2.39x10⁻⁵], cholelithiasis [P = 1.97x10⁻⁵], and cholecystitis [P = 6.54x10⁻⁷]), and non-specific clinical symptoms (hematuria [P = 1.14x10⁻⁵] and abdominal pain [P = 3.99x10⁻⁹]))(**Fig 1C** and **S3 Table**).

The two phenome-wide significant associations in MGI were morbid obesity within the endocrine/metabolic domain (P = 7.07x10⁻⁶) and acute sinusitis in the respiratory category

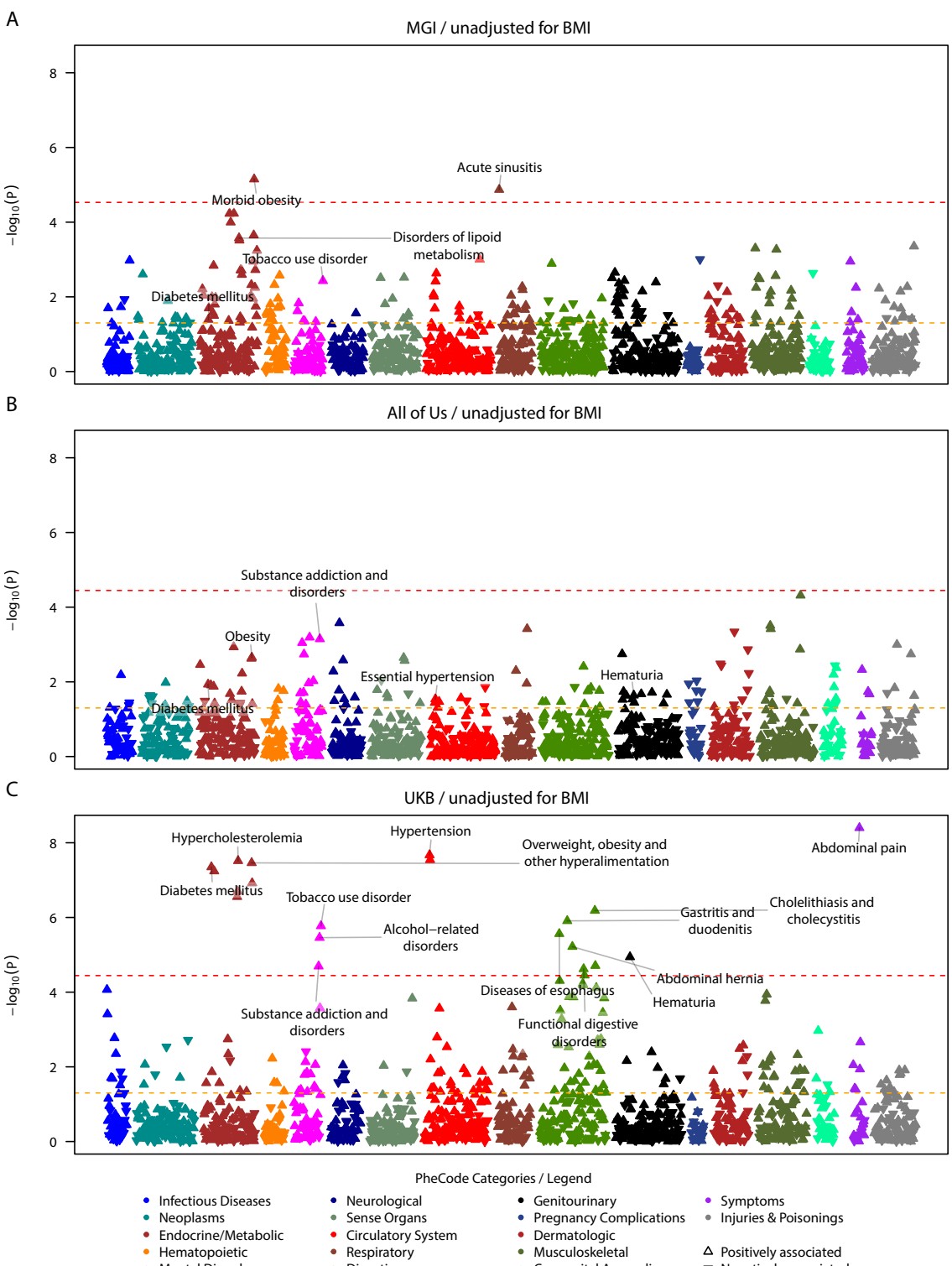

**Fig 1. Study-specific COVID-19 severity PRS PheWAS on pre-pandemic conditions in EUR individuals, unadjusted for BMI.** Study-specific PRS PheWAS results for MGI (top, 1694 PheCodes), All of Us (center, 1397 PheCodes), and UK Biobank (bottom, 1388 PheCodes) are shown. PheCodes are only labeled if they have reached nominal significance in one study and phenome-wide significance in another. To avoid overcrowding in the plot, for parent-sibling PheCode combinations, only the top PheCode is labeled. Summary statistics can be found in **S3 Table**. The dashed red line indicates the phenome-wide significance threshold, and the dashed yellow line

indicates the nominal significance threshold. The upward/downward orientation of the triangles indicates the positive/negative direction of the estimated association.

(P = 1.35x10$^{-5}$, **Fig 1A** and **S3 Table**). In MGI, among the phenome-wide significant hits of the UKB, disorders of lipoid metabolism (incl. hyperlipidemia and hypercholesterolemia) and tobacco use disorder reached nominal significance. Acute sinusitis reached phenome-wide/nominal significance solely in MGI.

While there were no phenome-wide significant hits observed in All of Us, some of the hits seen in the UKB analysis were positively and nominally associated with the COVID-19 severity PRS: substance addiction disorders (P = 7.14x10$^{-4}$), obesity (P = 0.0022), overweight, obesity and other hyperalimentation (P = 0.0024), diabetes mellitus (P = 0.012), type 2 diabetes (P = 0.012), hematuria (P = 0.023), essential hypertension (P = 0.029), hypertension (P = 0.036), and morbid obesity (P = 0.041)(**Fig 1B** and **S3 Table**).

To further investigate if the disorders of lipoid metabolism and hypertension, phenotypes known to be associated with obesity-related traits, were associated with the COVID-19 Severity PRS independently of BMI, we performed PheWAS adjusted for BMI. Again, we observed heterogeneity in the association results. Again, most associations were exclusively identified with phenome-wide significance in the UKB cohort: hypercholesterolemia, substance use disorders, gastrointestinal disorders, and hematuria remained phenome-wide significant. Other signals of metabolic and cardiovascular disorders (e.g., hypertension) were attenuated yet remained nominally significant (**Fig 2C** and **S4 Table**).

## Meta-analysis across three biobanks

We next conducted meta-analyses across the three cohorts to boost the power to detect associations in a combined sample size of 520,445. In the meta-analysis of the BMI-unadjusted PheWAS, we identified 27 phenome-wide significant phecodes (**Fig 3A** and **S9 Table**). Among them, five associations were unrelated to previously associated phenotypes. They belonged to respiratory, digestive, and musculoskeletal categories, such as chronic airway obstruction (P = 7.21 x 10$^{-6}$), diverticulosis and diverticulitis (P = 1.10 x 10$^{-5}$), diverticulosis (P = 1.30 x 10$^{-5}$), arthropathy NOS (P = 6.42 x 10$^{-6}$), and other arthropathies (P = 3.65 x 10$^{-6}$). When we meta-analyzed the BMI-adjusted PheWAS, 11 out of the initial 27 phenome-wide significant hits remained significant (**Figs 3B, S7** and **S10 Table,**). The remaining significant associations were observed in various categories, such as endocrine/metabolic (e.g., hypercholesterolemia [P = 8.24 x 10$^{-6}$]), mental disorders (e.g., tobacco use disorder [P = 6.80 x 10$^{-9}$]), respiratory (chronic airway obstruction [P = 2.83 x 10$^{-5}$]), digestive (gastritis and duodenitis [P = 9.04 x 10$^{-6}$]), genitourinary (hematuria [P = 1.23 x 10$^{-5}$]), and non-specific clinical symptoms (abdominal pain [P = 4.18 x 10$^{-8}$]). The heterogeneity in effect sizes across studies was generally low, with some exceptions, such as disorders of lipoid metabolism, hyperlipidemia, and abdominal pain (I$^2$ = 64.2, 64.0, and 67.9% in the BMI-unadjusted analysis, where I$^2$ describes the percentage of variation that is due to heterogeneity rather than chance); however, none of the signals showed significant between-study variance (Q-statistic p > 0.05) (**S9** and **S10 Tables** and **S8 Fig**).

## PheWAS analyses in African and East Asian ancestry groups

We also conducted PheWAS analyses in the AFR and EAS ancestry groups for each cohort, both with and without BMI adjustment (**S5–S8 Tables**). However, these analyses faced limitations due to low sample sizes, resulting in fewer phenotypes with over 50 cases and reduced

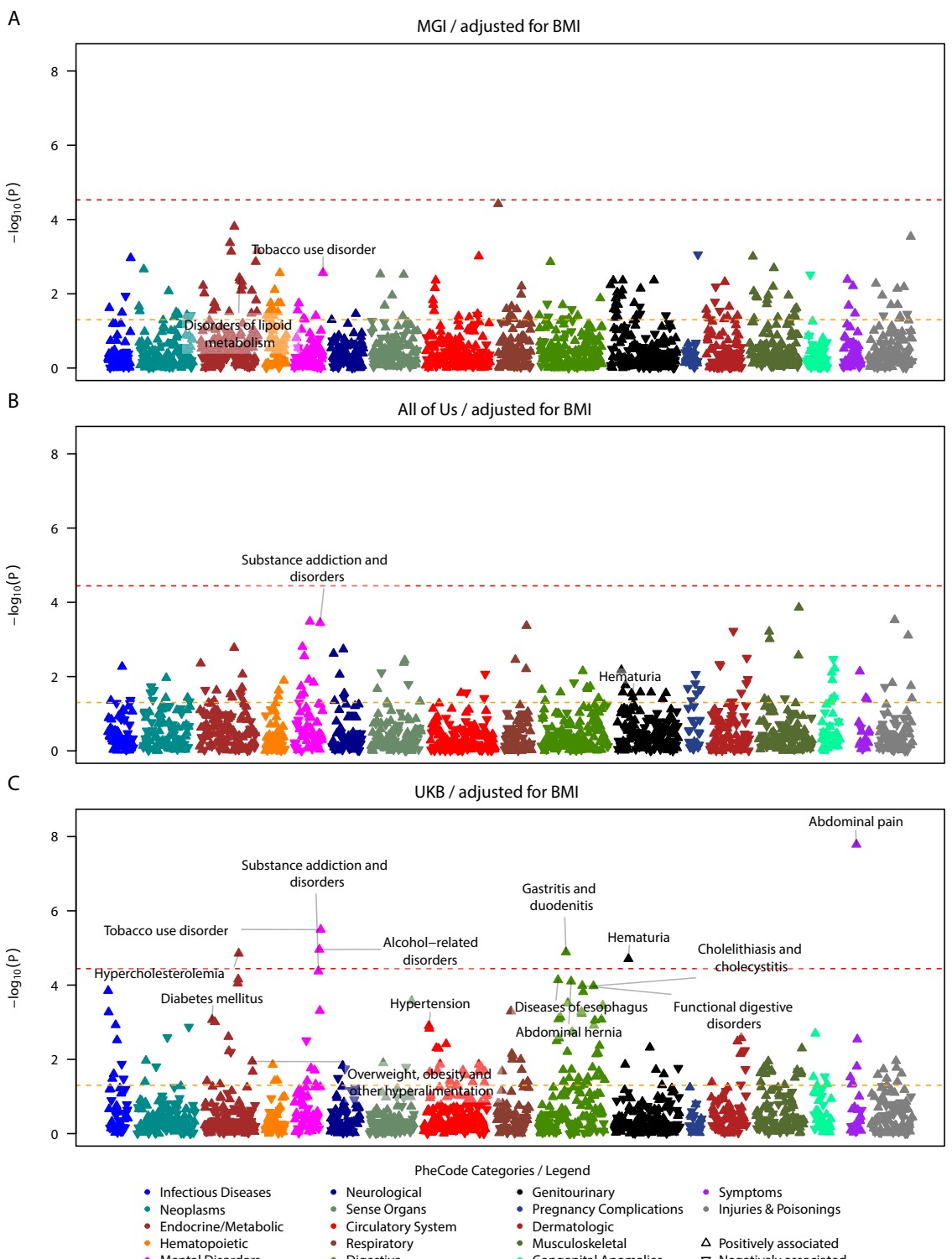

**Fig 2. Study specific COVID-19 severity PRS PheWAS on pre-pandemic conditions in EUR individuals, adjusted for BMI.** Study-specific PRS PheWAS results for MGI (top, 1694 PheCodes), All of Us (center, 1397 PheCodes), and UK Biobank (bottom, 1388 PheCodes) are shown. PheCodes are only labeled if they have reached nominal significance in one study and phenome-wide significance in another. To avoid overcrowding in the plot, for parent-sibling PheCode combinations, only the top PheCode is labeled. Summary statistics can be found in **S4 Table**. The dashed red line indicates the phenome-wide significance threshold, and the dashed yellow line

indicates the nominal significance threshold. The upward/downward orientation of the triangles indicates the positive/negative direction of the estimated association.

statistical power (see **S2 Table**). In the AFR ancestry group of the All of Us cohort, we identified 70 phenome-wide significant associations, which increased to 84 when adjusting for BMI. Some of the associated phenotypes overlapped with phenotypes identified in the EUR ancestry group: disorders of lipoid metabolism, hyperlipidemia, hypercholesterolemia, diseases of the esophagus, and gastritis and duodenitis (**S3 and S5 Tables**). However, these associations were negatively correlated, meaning that higher COVID-19 Severity PRS values were associated with lower risks of the respective phenotypes in the AFR ancestry group of All of Us. This finding contrasts with the positive associations observed in the EUR PheWAS analyses, where higher PRS values were associated with higher risks of the respective phenotypes. No associations were observed for the AFR ancestry group in MGI or UKB, nor for the EAS ancestry across any cohort. A subsequent meta-analysis of the PheWAS results revealed significant between-study heterogeneity, further complicating interpretation (**S11–S14 Tables**). These inconsistencies highlight the challenges in comparing non-EUR ancestry groups across the three cohorts and underscore the limitations of a PRS predominantly based on EUR individuals when applied to other ancestry groups.

## Supplementary analysis with population-based COVID-19 Severity PRS

In our main analyses, we used a PRS generated using a COVID-19 severity GWAS conditional on testing positive for COVID-19 to study which pre-existing conditions are associated with an increased predicted risk for hospitalization when infected with COVID-19. To complement these explorations, we also considered a PRS based on the hospitalized COVID-19 versus population GWAS ("B2 All") whose PRS PheWAS may provide additional insights into pre-existing conditions that are associated with an increased predicted risk for contracting COVID-19 that requires hospitalization.

The effective sample size of "B2 All" GWAS was more than 3 times larger than the "B1 All" GWAS; however, their case definition was identical, and their control definition overlapped, so not surprisingly, a correlation analysis of the resulting PRS in MGI revealed that both PRS are highly correlated in MGI (R = 0.474; **S3 Fig**).

The subsequent cohort-specific PheWAS and their meta-analysis with the B2_ALL PRS revealed top hits that largely aligned with the rankings of top signals from the B1_All PRS analyses. For instance, obesity, hypertension, tobacco use disorder, and abdominal pain consistently featured among the leading signals (**S4–S6 Figs, S14–S18 Tables**). Nevertheless, numerous additional phenotypes reached phenome-wide significance in the meta-analysis (unadjusted for BMI: 168 vs. 27; adjusted for BMI: 91 vs. 11 phenome-wide significant hits, **S19 Table** and **Figs 3 and S6**). In addition, while the ranking of the signals was largely consistent, they exhibited substantially lower P-values (e.g., obesity: $P_{B1\_ALL/BMI-unadjusted} = 1.9 \times 10^{-12}$ vs. $P_{B2\_ALL/BMI-unadjusted} = 1.2 \times 10^{-55}$; or tobacco use disorder: $P_{B1\_ALL/BMI-adjusted} = 6.8 \times 10^{-9}$ vs. $P_{B2\_ALL/BMI-adjusted} = 2.0 \times 10^{-22}$; **S19 Table** and **Figs 3 and S6**)

## Discussion

This study analyzed the associations between genetically predicted COVID-19 severity and pre-existing conditions captured in medical phenomes across three large biobanks: MGI, UKB, and All of Us, conducting cohort-specific PheWAS and a subsequent meta-analysis. Using a GWAS on COVID-19 hospitalization to create a PRS as a genetic proxy for COVID-

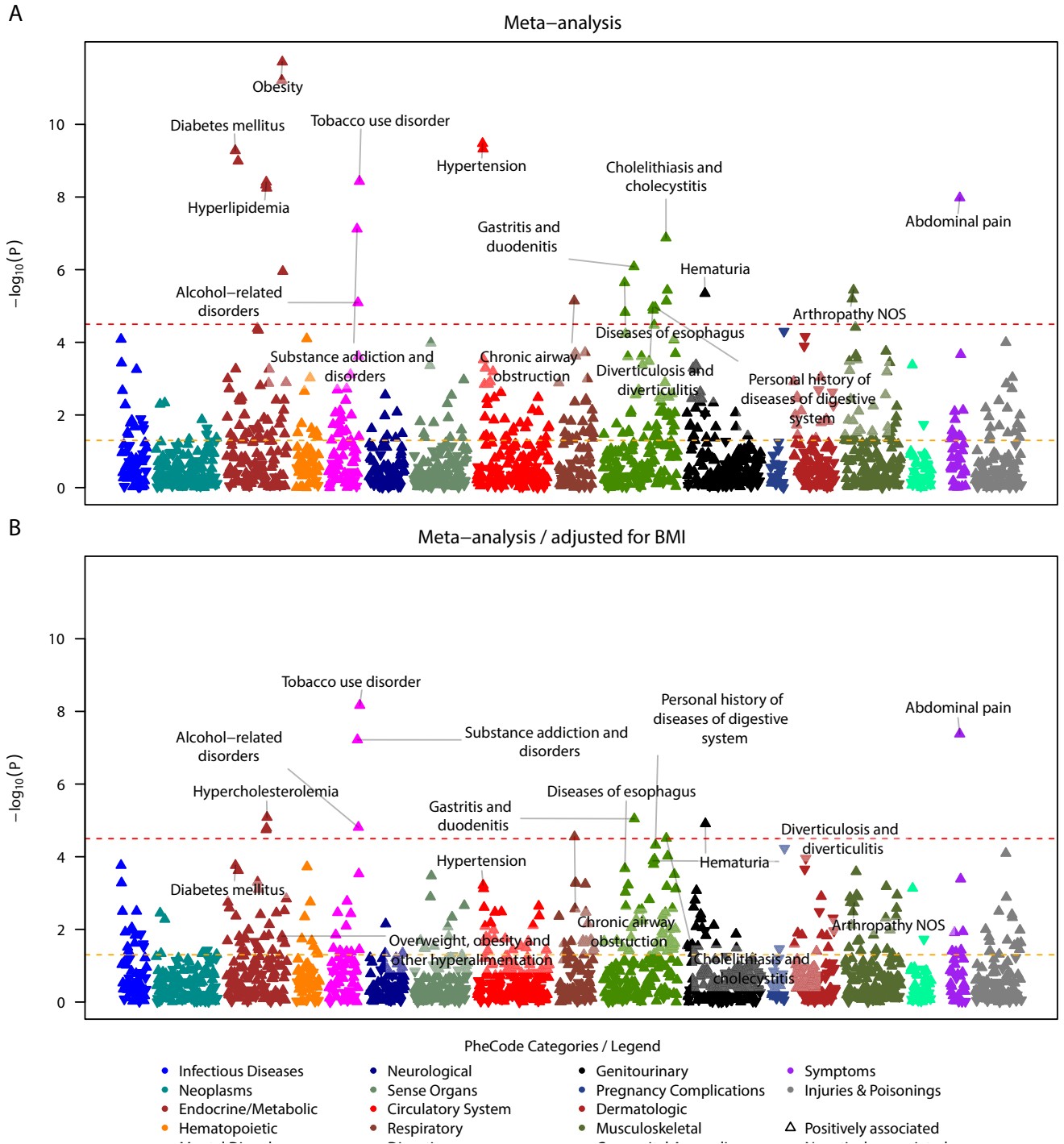

**Fig 3. Meta-analyzed COVID-19 severity PRS PheWAS on pre-pandemic conditions in EUR individuals.** A total of 1576 PheCodes that were analyzed in at least two studies are shown without (top) and with adjustment for BMI (bottom). PheCodes are only labeled if they have reached nominal significance in one analysis and phenome-wide significance in another. To avoid overcrowding in the plot, for parent-sibling PheCode combinations, only the top PheCode is labeled. Meta-analysis summary statistics can be found in **S9 and S10 Tables**. The dashed red line indicates the phenome-wide significance threshold, and the dashed yellow line indicates the nominal significance threshold. The upward/downward orientation of the triangles indicates the positive/negative direction of the estimated association.

19 severity, our approach allows for a more consistent and unbiased assessment of the associations between pre-existing conditions and COVID-19 severity across different cohorts.

The choice between the "B1" PRS and the "B2" PRS in our study hinged on the difference in the control groups used in their underlying GWAS. The "B1 All" GWAS used non-hospitalized COVID-19 cases as controls, while the "B2 All" GWAS utilized population controls comprising SARS-CoV-2 negative or untested individuals, which may include mildly symptomatic or asymptomatic individuals. Consequently, the "B1" PRS we used in our main analysis might arguably be more suitable to capture the heritable risk for COVID-19 severity, while a "B2" PRS would capture the heritable risk for COVID-19 susceptibility.

In the biobank-specific analyses, we identified 23 pre-existing conditions significantly associated with the COVID-19 PRS ($p<3.6x10^{-5}$), with 21 observed in the UKB cohort and two in the MGI cohort. The meta-analysis uncovered 27 significant phenotypes ($p<3.6x10^{-5}$), predominantly related to obesity, metabolic disorders, and cardiovascular conditions. Our findings expand upon the growing evidence of the complex interplay between clinical phenotypes and COVID-19 outcomes, confirming mostly known factors and enhancing our understanding of the relationship between pre-existing clinical phenotypes and COVID-19 severity.

To follow up on our findings, we performed Mendelian Randomization (MR) analyses and applied a range of statistical methods, including MR Egger, Weighted Median, Inverse Variance Weighted (IVW), Simple Mode, and Weighted Mode, to assess the genetic evidence for a causal relationship between smoking-related traits and COVID-19 outcomes (**S1 Text**). For both smoking initiation and cigarettes per day [62], the majority of MR analyses did not demonstrate significant causation with COVID-19 severity (B1) or susceptibility (B2), with p-values generally exceeding the nominal significance threshold, indicating no robust genetic causal effect. For the B2 outcome, the IVW method yielded a marginally significant p-value (p=0.023) for the number of cigarettes smoked per day, a finding corroborated by the MR-PRESSO (p=0.025 pre-outlier correction; p=0.029 post-outlier correction; **S10–S13 Figs and S20 Table**) [63–67]. The observed effect sizes were relatively small, suggesting only weak evidence of causality between cigarette consumption and increased COVID-19 susceptibility. While the MR approach strengthens the causal inferences that can be drawn from our study, we recognize that these findings are inconclusive and require follow-up with well-powered studies to understand the implications of our results fully.

Noteworthy, our COVID-19 severity PRS PheWAS findings within the UK Biobank cohort showed striking similarities to our previously published COVID-19-unrelated BMI PRS PheWAS results [35]. Given that the underlying GWAS on COVID-19 severity did not control for BMI, it is plausible that the COVID-19 severity PRS indirectly captured part of the genetic predisposition for overweight and obesity, a major risk factor for COVID-19 complications [68–71]. By performing a second set of PheWAS with adjustment for BMI, we could discern several clinical phenotypes, e.g., hypercholesterolemia and gastrointestinal disorders, that, independent of high BMI (i.e., BMI $\geq$ 25), may heighten the risk of hospitalization following COVID-19 infection.

In examining the distribution of diagnoses across various categories of diseases in unrelated European ancestry cohorts from hospital-based (MGI), population-based (UKB), and the All of Us cohorts, certain patterns emerge, as illustrated in **S14 Fig**. Generally, the MGI cohort exhibits a higher proportion of affected individuals across all categories, reflective of its hospital-based nature [42]. In contrast, the UKB data, representing a population-based sample, consistently reports lower diagnosis rates, especially for congenital anomalies [72]. The All of Us cohort demonstrates intermediate values reflective of their recruitment, a mix of open invitations and partnerships with healthcare provider organizations [44]. These observations highlight the variability in health condition prevalence across different cohort types and

underscore the importance of considering the cohort source and recruitment strategies when interpreting disease frequency data.

In all three biobanks, the outcome "hospitalization due to COVID-19" was relatively sparse (MGI: $n_{Hospitalized}$ = 307, UKB: $n_{Hospitalized}$ = 3165, All of Us: $n_{Hospitalized}$ = 703, **Table 1**) and thus would have limited the statistical power of the study. In contrast, a PRS for COVID-19 severity allowed us to detect significant associations between pre-existing conditions and COVID-19 severity in substantially larger datasets (MGI: n = 47,257, UKB: n = 425,787, All of Us: n = 47,401), even though the PRS might capture only a smaller fraction of the trait's variability. To illustrate, the limited sample size in the three cohorts precluded us from conducting formal performance analyses and enrichment analyses when we attempted to evaluate the potential of the COVID-19 severity PRS to stratify hospitalized cases. When examining the number of hospitalized COVID-19 cases across PRS quartiles in the analytical datasets (**Table 2**), no clear trend emerged within the MGI cohort, with case numbers ranging between 48 and 56, close to the average of 54 cases. In the UKB cohort, we observed a marked trend between the lowest quartile (n = 522) and the highest quartile (n = 735). In contrast, the central quartiles (Q2: n = 670 and Q3: n = 633) exhibited case numbers close to the average count (n = 640). In the All of Us cohort, the numbers of the upper half were substantially higher (Q3: n = 102 and Q4: n = 83) than the average of 74 cases and the lower half (Q1: n = 53 and Q2: n = 60, **S9 Fig**). Identifying non-hospitalized COVID-19 cases in the three biobanks was not feasible because primary care data was limited or not available/accessible.

Despite its limitations, the PRS approach provided a powerful alternative to actual COVID-19 outcomes because it only requires external GWAS summary statistics and pre-COVID-19 EHR data highlighting its potential value for studies where actual outcome data may be unavailable or insufficient for robust analyses.

A previous study employing UK Biobank data constructed an 86-SNP PRS for COVID-19 severity, based on the HGI Consortium's GWAS results on "very severe respiratory confirmed COVID-19 vs. population" [33]. The study found that including the PRS in a basic prediction model (considering only sex, age, and income) improved the AUC by a modest 0.3%. So, while this study's definition of COVID-19 severity, as "very severe respiratory confirmed COVID-19", diverges from our hospitalization-based definition, it highlights the lack of predictive power through a severity COVID-19 PRS. Nevertheless, the study showed that adding pre-COVID-19 diagnoses, quantified as Charlson Comorbidity Index Scores, yielded the largest model improvement, emphasizing the importance of characterizing pre-COVID-19 diagnoses for identifying at-risk individuals.

As mentioned above, we consciously opted to use the "B1 All" PRS for our main analyses, as it was conditional on COVID-19 cases centering on the genetic determinants of COVID-19 severity post-infection instead of on the genetic determinants of COVID-19 susceptibility and severity. However, we also analyzed the corresponding "B2 All" PRS to provide a more comprehensive perspective on pre-existing conditions associated with genetic predispositions related to COVID-19. A comparison of both PRSs indicated a strong correlation and consistently top-ranked associations, confirming the previously reported overlap between identified susceptibility and severity risk variants [30,73]. The "B2 All" GWAS approach, with its more lenient control definition, allows larger sample sizes and thus might enable more powerful discovery of COVID-19-related risk variants. However, further follow-up studies will be needed to understand whether these risk variants are informative regarding susceptibility/protection, severity/expedited treatment, or both.

Our study's application of a comprehensive GWAS-derived PRS for PheWAS is novel in COVID-19, mirroring successful strategies in other genomic research areas [74–76]. Unlike most studies where COVID-19 risk SNPs have been used as weak instruments [77], in our

research, a PRS serves as a robust proxy for the severity of COVID-19. This is particularly significant given the availability of accurate PRS data for a large number of individuals, contrasting with the often poor quality or incompleteness of COVID-19 outcome data. By broadly capturing genetic variations related to COVID-19 outcomes, our PRS expands risk prediction capabilities beyond what is possible with analyses restricted to known loci like *ACE2* and *TMPRSS2*. This agnostic approach aligns with the aims of initiatives such as the COVID-19 Host Genetics Initiative, which seeks to discover genetic factors impacting patient outcomes [78], underscoring the value of wide-ranging genetic investigations in understanding disease risks and informing clinical decisions.

Our study presents several limitations, which should be considered when interpreting our findings. Firstly, the inconsistent measurement of COVID-19 severity across cohorts poses a challenge, as our study utilized hospitalization as the criteria for severity. At the same time, other investigations may define it based on COVID-19-related acute respiratory distress syndrome, multi-organ failure, or death [19,20]. This discrepancy complicates comparisons with other studies, potentially limiting the generalizability and applicability of our results. Future research should strive for a more standardized definition of COVID-19 severity, incorporating additional proxies such as mechanical ventilation requirements, ICU admissions, or specific immune biomarkers, to improve the evaluation of severity PRS models and facilitate cross-study comparisons.

Secondly, we did not assess the predictive performance of the COVID-19 severity PRS as it is usually recommended for newly developed PRS [79] due to a lack of well-characterized COVID-19 cases/severity and small sample sizes. Instead, we relied only on the discovery of GWAS and the applied PRS method, i.e., any biases or confounding in the underlying GWAS may have also biased the resulting PRS. In particular, the predictive accuracy of PRS is likely diminished for non-European individuals due to the GWAS being based primarily on European samples, where EUR-specific environmental and socio-economic factors, in addition to genetic factors, may significantly influence COVID-19 severity.

Thirdly, our approach did not work for non-European subsets, which could be due to their substantially smaller sample sizes and the well-established lack of transportability of PRS across diverse populations [80]. This underscores the need to establish larger, more diverse populations, particularly by including more representative non-European samples and applying ancestry-aware PRS methods, thereby enhancing the accuracy and broader applicability of COVID-19 severity PRS investigations across diverse ethnic groups.

Finally, we did not account for selection bias in the three cohorts, which could explain some of the heterogeneity we observed in the meta-analysis. For example, MGI is a hospital-based cohort enriched for patients undergoing surgery [42], and UKB is a population-based cohort that was reported to have a "healthy volunteer" selection bias [72]. At the same time, All of Us has purposefully oversampled certain underrepresented subgroups [44,81]. While many of our PheWAS results align with previous reports, moving forward, it is imperative to include and analyze more representative samples of non-European populations and to apply ancestry-aware PRS methods to improve the accuracy and applicability of PRS PheWAS in diverse ancestry groups.

Given that PRS for a range of traits will be accessible for every genotyped individual – a major advantage over missing EHR data – it may be advantageous to explore the development of a multi-trait PRS approach [35]. Such an approach could offer a more comprehensive and precise risk evaluation, allowing for improved targeting of interventions and resource distribution while circumventing potential confounding or data incompleteness in existing health records. Recent investigations have showcased the feasibility and possible advantages of multi-trait PRS in alternative settings, such as cardiovascular disease risk prediction [82]. By

combining risk factors (or their genetic proxies) that contribute to COVID-19 severity, we could achieve a more comprehensive understanding of individual risk, thereby enabling the implementation of more personalized public health strategies.

## Conclusion

Our study investigated the associations between COVID-19 severity and pre-existing conditions across three large biobanks, using a PRS as a proxy for COVID-19 severity to circumvent challenges commonly found in EHR data. These challenges include sparsity of COVID-19 severity data, missingness, biases, and misclassification – when hospitalization cannot be definitively attributed to COVID-19. Our findings revealed significant associations with obesity, metabolic disorders, and cardiovascular conditions, confirming known factors and expanding our understanding of the relationship between pre-existing clinical phenotypes and COVID-19 outcomes.

## Web resources

Michigan Genomics Initiative (MGI): https://precisionhealth.umich.edu/our-research/michigangenomics/

PLINK 2.0, https://www.cog-genomics.org/plink/2.0/

UK Biobank dataset, https://www.ebi.ac.uk/ega/datasets/EGAD00010001474

FRAPOSA, https://github.com/daviddaiweizhang/fraposa

All of Us cohort, https://databrowser.researchallofus.org/

COVID-19 Host Genetics Initiative (COVID19-hg GWAS meta-analyses round 7), https://www.covid19hg.org/results/r7/

PRS-CS, https://github.com/getian107/PRScs

PRS Weights: https://csg.sph.umich.edu/larsf/SuppData/COVID19_PRS_2023/

Smoking GWAS summary statistics: https://conservancy.umn.edu/handle/11299/201564

## Supporting information

**S1 Text. Supplementary Methods (incl. references [62–67]).**
(DOCX)

**S1 Fig. Scatter plots illustrating PRS and BMI relationships in the MGI cohort.**
(DOCX)

**S2 Fig. Scatter plot depicting the correlation between BMI and COVID-19 severity GWAS effect sizes.**
(DOCX)

**S3 Fig. Scatter plot depicting the correlation between the "B1_ALL" COVID-19 Severity PRS and the "B2_ALL" COVID-19 Severity PRS in the MGI cohort.**
(DOCX)

**S4 Fig. Study-specific COVID-19 severity "B2_ALL" PRS PheWAS on pre-pandemic conditions in EUR individuals, unadjusted for BMI.**
(DOCX)

**S5 Fig. Study specific COVID-19 severity "B2_ALL" PRS PheWAS on pre-pandemic conditions in EUR individuals, adjusted for BMI.**
(DOCX)

**S6 Fig. Meta-analyzed COVID-19 severity "B2_ALL" PRS PheWAS on pre-pandemic conditions in EUR individuals.**
(DOCX)

**S7 Fig. Forest plots of the BMI-unadjusted association between the PRS for COVID-19 severity and various pre-pandemic phenotypes.**
(DOCX)

**S8 Fig. Forest plots of the BMI-adjusted association between the PRS for COVID-19 severity and various pre-pandemic phenotypes.**
(DOCX)

**S9 Fig. COVID-19 hospitalizations in PRS quartiles of European ancestry individuals of the three cohorts.**
(DOCX)

**S10 Fig. Scatter plot demonstrating the SNP effect of smoking initiation on COVID-19 severity.**
(DOCX)

**S11 Fig. Scatter plot demonstrating the SNP effect of cigarettes smoked per day on COVID-19 severity.**
(DOCX)

**S12 Fig. Scatter plot demonstrating the SNP effect of smoking initiation on COVID-19 susceptibility.**
(DOCX)

**S13 Fig. Scatter plot demonstrating the SNP effect of cigarettes smoked per day on COVID-19 susceptibility.**
(DOCX)

**S14 Fig. Prevalence of Phenotype Categories in the three analytical datasets.**
(DOCX)

**S1 Table. Sample Sizes and Ancestry of Studies in COVID-19 HGI GWAS Meta-Analysis "B1_ALL".**
(XLSX)

**S2 Table. Number of PheCodes by Analysis (Ancestry EUR, AFR, or EAS; Unadjusted for BMI or Adjusted for BMI).**
(XLSX)

**S3 Table. Cohort-specific "B1_ALL" PRS PheWAS Results for EUR Ancestry, Unadjusted for BMI.**
(XLSX)

**S4 Table. Cohort-specific "B1_ALL" PRS PheWAS Results for EUR Ancestry, Adjusted for BMI.**
(XLSX)

**S5 Table. Cohort-specific "B1_ALL" PRS PheWAS Results for AFR Ancestry, Unadjusted for BMI.**
(XLSX)

**S6 Table. Cohort-specific "B1_ALL" PRS PheWAS Results for AFR Ancestry, Adjusted for BMI.**
(XLSX)

**S7 Table. Cohort-specific "B1_ALL" PRS PheWAS Results for EAS Ancestry, Unadjusted for BMI.**
(XLSX)

**S8 Table. Cohort-specific "B1_ALL" PRS PheWAS Results for EAS Ancestry, Adjusted for BMI.**
(XLSX)

**S9 Table. B1_ALL" PRS PheWAS Meta-Analysis Results for EUR Ancestry, Unadjusted for BMI.**
(XLSX)

**S10 Table. "B1_ALL" PRS PheWAS Meta-Analysis Results for EUR Ancestry, Adjusted for BMI.**
(XLSX)

**S11 Table. B1_ALL" PRS PheWAS Meta-Analysis Results for AFR Ancestry, Unadjusted for BMI.**
(XLSX)

**S12 Table. "B1_ALL" PRS PheWAS Meta-Analysis Results for AFR Ancestry, Adjusted for BMI.**
(XLSX)

**S13 Table. "B1_ALL" PRS PheWAS Meta-Analysis Results for EAS Ancestry, Unadjusted for BMI.**
(XLSX)

**S14 Table. B1_ALL" PRS PheWAS Meta-Analysis Results for EAS Ancestry, Adjusted for BMI.**
(XLSX)

**S15 Table. Cohort-specific "B2_ALL" PRS PheWAS Results for EUR Ancestry, Unadjusted for BMI.**
(XLSX)

**S16 Table. Cohort-specific "B2_ALL" PRS PheWAS Results for EUR Ancestry, Adjusted for BMI.**
(XLSX)

**S17 Table. "B2_ALL" PRS PheWAS Meta-Analysis Results for EUR Ancestry, Unadjusted for BMI.**
(XLSX)

**S18 Table. B2_ALL" PRS PheWAS Meta-Analysis Results for EUR Ancestry, Adjusted for BMI.**
(XLSX)

**S19 Table. Summary of "B1_ALL" and "B2_ALL" PRS PheWAS Meta-Analysis Results for EUR Ancestry.**
(XLSX)

**S20 Table. Mendelian randomization analysis for the effect of smoking initiation or cigarettes consumed per day on COVID-19 severity or COVID-19 susceptibility.**
(XLSX)

## Acknowledgments

The authors acknowledge the Michigan Genomics Initiative participants, Precision Health at the University of Michigan, the University of Michigan Medical School Central Biorepository, the University of Michigan Medical School Data Office for Clinical and Translational Research, and the University of Michigan Advanced Genomics Core for providing data and specimen storage, management, processing, and distribution services, and the Center for Statistical Genetics in the Department of Biostatistics at the School of Public Health for genotype data curation, imputation, and management in support of the research reported in this publication/grant application/presentation.

This research has been conducted using the UK Biobank Resource under application number 24460. The All of Us Research Program is supported by the National Institutes of Health, Office of the Director: Regional Medical Centers: 1 OT2 OD026549; 1 OT2 OD026554; 1 OT2 OD026557; 1 OT2 OD026556; 1 OT2 OD026550; 1 OT2 OD 026552; 1 OT2 OD026553; 1 OT2 OD026548; 1 OT2 OD026551; 1 OT2 OD026555; IAA #: AOD 16037; Federally Qualified Health Centers: HHSN 263201600085U; Data and Research Center: 5 U2C OD023196; Biobank: 1 U24 OD023121; The Participant Center: U24 OD023176; Participant Technology Systems Center: 1 U24 OD023163; Communications and Engagement: 3 OT2 OD023205; 3 OT2 OD023206; and Community Partners: 1 OT2 OD025277; 3 OT2 OD025315; 1 OT2 OD025337; 1 OT2 OD025276. In addition, the All of Us Research Program would not be possible without the partnership of its participants

## Author Contributions

**Conceptualization:** Lars G. Fritsche, Bhramar Mukherjee.

**Data curation:** Lars G. Fritsche.

**Formal analysis:** Lars G. Fritsche, Kisung Nam.

**Funding acquisition:** Bhramar Mukherjee.

**Investigation:** Lars G. Fritsche.

**Methodology:** Lars G. Fritsche, Bhramar Mukherjee.

**Project administration:** Lars G. Fritsche, Bhramar Mukherjee.

**Resources:** Lars G. Fritsche, Bhramar Mukherjee.

**Software:** Lars G. Fritsche, Kisung Nam.

**Supervision:** Lars G. Fritsche, Seunggeun Lee, Bhramar Mukherjee.

**Visualization:** Lars G. Fritsche, Kisung Nam.

**Writing – original draft:** Lars G. Fritsche, Jiacong Du, Bhramar Mukherjee.

**Writing – review & editing:** Lars G. Fritsche, Kisung Nam, Jiacong Du, Ritoban Kundu, Maxwell Salvatore, Xu Shi, Seunggeun Lee, Stephen Burgess, Bhramar Mukherjee.

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
