## [Decision Letter · Decision Letter 0]

13 Sep 2023

Dear Dr Fritsche,

Thank you very much for submitting your Research Article entitled 'Uncovering Associations between Pre-existing Conditions and COVID-19 Severity: A Polygenic Risk Score Approach Across Three Large Biobanks' to PLOS Genetics.

The manuscript was fully evaluated at the editorial level and by independent peer reviewers. The reviewers appreciated the attention to an important problem, but raised some substantial concerns about the current manuscript. Based on the reviews, we will not be able to accept this version of the manuscript, but we would be willing to review a much-revised version. We cannot, of course, promise publication at that time.

If you decide to revise the manuscript for further consideration at PLOS Genetics, please aim to resubmit within the next 60 days, unless it will take extra time to address the concerns of the reviewers, in which case we would appreciate an expected resubmission date by email to plosgenetics@plos.org.

We are sorry that we cannot be more positive about your manuscript at this stage. Please do not hesitate to contact us if you have any concerns or questions.

Yours sincerely,

Giorgio Sirugo

Academic Editor

PLOS Genetics

Hua Tang

Section Editor

PLOS Genetics

Reviewer's Responses to Questions

**Comments to the Authors:**

Reviewer #1: The authors used PRS based on COVID GWAS as a proxy of severity of COVID and performed pheWAS in EUR samples from 3 biobanks (UKBB, MGI, and All of US) . They found significant association between the PRS and phenotypes related with obesity, metabolic disorders, and cardiovascular conditions and the signal mainly came from UKBB EUR samples. In addition to the analysis performed in EUR samples, the authors also performed pheWAS in non-EUR samples but the signals were either non-significant or negatively correlated with the findings in EUR samples.

While the scholarly pursuit undertaken is undeniably of substantial pertinence, there exist prospects for further enhancement of its methodological robustness.

First, COVID GWAS contains UKBB EUR samples. The samples that consist of one third of the whole discovery GWAS sample size were then used as the target data in the pheWAS analysis, and most of the signals are from UKBB EUR samples. If the phenotype used in the discovery GWAS and the target data prediction are the same, the overfitting would be very severe for sure. Although the phenotype used in discovery GWAS and pheWAS are different, they still can be correlated and therefore cause overfitting pheWAS results. The authors should rule out the possibility that the signals from the UKBB EUR sample result from overfitting, especially when almost all the significant results are from the UKBB EUR samples. Since the pheWAS in EUR samples is the main findings of this project, all the following analysis and discussion would be questionable if validity of this result cannot be confirmed. One solution could be collaborating with COVID GWAS consortium and getting a GWAS without UKBB samples.

Second, I agree with the authors that the inconsistency between EUR and non-EUR results can be caused by the small sample size of non-EUR data and low transferability across populations. I would like to further point out that for traits that are influenced by many confounding factors, like getting COVID during the pandemic, the PRS transferability across populations can be even lower than traits mainly caused by biological or genetic factors . The COVID GWAS used in this study is mainly based on EUR samples. The PRS based on this GWAS is very likely to be heavily influenced by EUR-specific factors (not only LD structure, but also other confounding factors only existing in the EUR populations) and therefore cannot sufficiently represent the likelihood of getting COVID or the severity of COVID in non-EUR populations. It could be another reason for signals from the AFR and EUR samples being negatively correlated.

I also agree with the authors that the correlation between the PRS and the actual severity of COVID could not be tested with the current data due to the existing poor phenotyping of COVID. Therefore, it would be great if the author could have other proxies of severity of COVID to support the findings based on the COVID PRS.

In conclusion, while the authors' work addressed an important topic of the relationships between COVID severity PRS and various phenotypic traits, the study would greatly benefit from addressing the aforementioned concerns to fortify the overall robustness and reliability of its findings.

Reviewer #2: Report: Uncovering Associations between Pre-existing Conditions and COVID-19 Severity: A Polygenic Risk Score Approach Across Three Large Biobanks

The main contribution of this paper is, Authors investigated the use of polygenic risk scores (PRS) as reliable proxies of COVID-19 severity across three large biobanks: the Michigan Genomics Initiative (MGI), UK Biobank (UKB), and NIH All of Us, to identify associations between pre-existing conditions and COVID-19 severity. By utilizing PRS as a proxy for COVID-19 severity, Authors identified known risk factors and novel associations between pre-existing clinical phenotypes and COVID-19 severity.

1. Authors performed the analysis stratified by Biobanks due to varying sampling strategies in these Biobanks. It would be great to include the details in the paper such as how they are not comparable with each other.

2. Would it be possible to describe the phenotypic categories in each bank separately with the help of a plot?

3. In addition to the table, it would be great to show the association of phenotypes, PheWAS results with the help of plot such as Forest plot.

4. Please include the details of the summary statistics on COVID-19 severity such the details of the participants, summary of the analysis etc.

5. Even though PRS-CS is a well-known PRS development technique, it would be good to include short summary of the algorithm in the paper.

6. Please include in the discussion section how this paper is different from similar papers in the literature such as if we use only the PRS with known loci.

Reviewer #3: The manuscript explores the role of genetic factors in determining the severity of COVID-19 outcomes, utilizing Polygenic Risk Scores (PRS) to predict COVID-19 severity. The authors analyzed genetic data from over half a million individuals across three large biobanks, aiming to identify individuals at high risk of severe illness due to COVID-19. By leveraging PRS, the study seeks to overcome challenges posed by data availability and quality, aiming to inform targeted interventions and prevention measures. The approach could potentially facilitate personalized healthcare, but I have few concerns:

- Causal Inference: The authors aim to shed light on the shared genetic susceptibility of COVID-19 severity and pre-existing conditions. However, establishing associations through PRS does not necessarily imply causality. The study would benefit from a Mendelian Randomization (MR) approach to infer causality and better inform intervention strategies.

- PRS Generation: The study exclusively utilized PRS-CS for generating PRS, overlooking the PRS-CSX method which is known to be more suitable for multiple ancestries. This choice might limit the generalizability of the findings across diverse populations.

- COVID Severity PRS Link with Tobacco Use Disorder: The manuscript highlighted a significant association between COVID-19 severity PRS and tobacco use disorder in both B1_ALL and B2_ALL analyses. This finding raises questions regarding the causal relationship between the two, warranting further investigation through MR to understand the underlying causality.

- The study presents an approach to understanding the genetic predispositions to COVID-19 severity, offering valuable insights into managing risks associated with the disease. However, it is imperative to address the limitations and to substantiate the findings through causal inference studies to enhance the robustness and applicability of the approach in real-world settings.

**Have all data underlying the figures and results presented in the manuscript been provided?**

Reviewer #1: Yes

Reviewer #2: Yes

Reviewer #3: Yes

PLOS authors have the option to publish the peer review history of their article (what does this mean?). If published, this will include your full peer review and any attached files.

Reviewer #1: No

Reviewer #2: No

Reviewer #3: **Yes: **Shefali Setia Verma

---

## [Decision Letter · Decision Letter 1]

5 Dec 2023

Dear Dr Fritsche,

We are pleased to inform you that your manuscript entitled "Uncovering Associations between Pre-existing Conditions and COVID-19 Severity: A Polygenic Risk Score Approach Across Three Large Biobanks" has been editorially accepted for publication in PLOS Genetics. Congratulations!

Yours sincerely,

Giorgio Sirugo

Academic Editor

PLOS Genetics

Hua Tang

Section Editor

PLOS Genetics

Comments from the reviewers (if applicable):

Since the Mendelian Randomization analyses did not produce significant results I strongly encourage the authors to underscore/elaborate on this limitation of their study.

Reviewer's Responses to Questions

**Comments to the Authors:**

Reviewer #1: I am satisfied with the author’s responses to my questions/issues raised in my initial review. I recommend that the revised paper be accepted

Reviewer #2: The authors were responsive to the comments and the manuscript has improved with the additional details and analysis.

Reviewer #3: Reviewers addressed all comments but note that MR analyses did not yield anything significant so it is important to not over sell the approach. More on MR in limitations section can be added to the paper

**Have all data underlying the figures and results presented in the manuscript been provided?**

Reviewer #1: Yes

Reviewer #2: Yes

Reviewer #3: Yes

PLOS authors have the option to publish the peer review history of their article (what does this mean?). If published, this will include your full peer review and any attached files.

Reviewer #1: No

Reviewer #2: No

Reviewer #3: No

**Data Deposition**

http://datadryad.org/submit?journalID=pgenetics&manu=PGENETICS-D-23-00893R1

**Press Queries**

---

## [Editor Report · Acceptance letter]

14 Dec 2023

PGENETICS-D-23-00893R1 

Uncovering Associations between Pre-existing Conditions and COVID-19 Severity: A Polygenic Risk Score Approach Across Three Large Biobanks 

Dear Dr Fritsche, 

We are pleased to inform you that your manuscript entitled "Uncovering Associations between Pre-existing Conditions and COVID-19 Severity: A Polygenic Risk Score Approach Across Three Large Biobanks" has been formally accepted for publication in PLOS Genetics! Your manuscript is now with our production department and you will be notified of the publication date in due course.

With kind regards,

Zsofi Zombor

PLOS Genetics

On behalf of:
